# Evaluation of Safety, Tolerability and Pharmacokinetic Characteristics of SA001 and Its Active Metabolite Rebamipide after Single and Multiple Oral Administration

**DOI:** 10.3390/ph16010132

**Published:** 2023-01-16

**Authors:** Sungyeun Bae, Ki Young Huh, Jaeseong Oh, Kyung-Sang Yu, Anhye Kim

**Affiliations:** 1Department of Clinical Pharmacology and Therapeutics, Seoul National University College of Medicine and Hospital, Seoul 03080, Republic of Korea; 2Department of Clinical Pharmacology and Therapeutics, CHA Bundang Medical Center, CHA University, Seongnam 13496, Republic of Korea; 3Department of Biomedical Informatics, CHA University School of Medicine, CHA University, Seongnam 13488, Republic of Korea; 4Institute for Biomedical Informatics, CHA University School of Medicine, CHA University, Seongnam 13488, Republic of Korea

**Keywords:** dry eye syndrome, Sjogren syndrome, phase 1 clinical trials, bioavailability

## Abstract

Dry eye disease (DED) is one of the most common eye diseases caused by multiple factors. Rebamipide, which is currently used to treat peptic ulcer disease, was shown to enhance secretory function and modulate inflammation in animal disease models. Considering the pathophysiology of DED, SA001 was developed expecting enhanced systemic exposure of rebamipide. Clinical trials to evaluate the safety, tolerability and pharmacokinetic (PK) characteristics of SA001 and its active metabolite rebamipide were conducted. After oral administration of SA001, blood and urine samples were collected for PK analysis of SA001 and rebamipide. PK parameters were compared between SA001 and conventional rebamipide (Bamedin^®^) and also between fasted and fed. Safety and tolerability were evaluated throughout the study based on adverse events (AEs), physical examinations, vital signs, 12-lead electrocardiography and clinical laboratory tests. SA001 was rapidly absorbed and quickly converted to rebamipide. The systemic exposure of rebamipide was dose-proportional after single and multiple doses. The plasma concentration of rebamipide after administration of SA001 was higher with a dose adjusted AUC_last_ and C_max_ 2.20 and 5.45 times higher in the 240 mg dose group and 4.73 and 11.94 times higher in the 600 mg dose group compared to conventional rebamipide. The favorable PK and tolerability profiles support further clinical development.

## 1. Introduction

Dry eye disease (DED) is one of the most common reasons for patients to seek ophthalmic care [1]. The prevalence of DED varies depending on the diagnostic criteria and study population and the population suffering from symptomatic DED is reported to range from 10 to 50% [2]. One of the reasons for such diversity is that there are various definitions for DED [3]. To make a global consensus, discussions were held by the American Academy of Ophthalmology, Asia Dry Eye Society and Tear Film and Ocular Surface Society Dry Eye Workshop. They concluded that “Dry eye is a multifactorial disease characterized by a persistently unstable and/or deficient tear film causing discomfort and/or visual impairment, accompanied by variable degrees of ocular surface epitheliopathy, inflammation and neurosensory abnormalities” [3]. Considering the multifactorial characteristics of DED, there are various treatment options from topical therapeutics such as ocular lubricants and cyclosporine A ophthalmic emulsion to systemic medications including tetracyclines and macrolides [4,5]. However, the management of DED is still controversial, and there is a clinically unmet need for a first-line treatment [6].

Rebamipide, known to stimulate prostaglandin and mucus glycoprotein synthesis, was originally developed by Otsuka Pharmaceutical Co., Ltd. (Tokyo, Japan) for the treatment of peptic ulcer disease [7]. Rebamipide was shown to enhance secretory function and modulate inflammation in animal disease models [8,9,10,11,12,13], and was considered a potential candidate for other diseases such as Sjogren’s syndrome and inflammatory bowel disease [14,15]. Rebamipide ophthalmic solution was both effective and tolerable in DED patients [16,17] and was developed as eye drops [18]. However, it needs to be administered every 4–6 h due to its limited therapeutic duration. Frequent administration of eye drops is associated with low patient compliance [19], and due to the discomfort, pain and foreign body sensation caused by the conventional ophthalmic solution, oral administration of rebamipide was considered as a potential substitute. However, the bioavailability of rebamipide was too low (10%) to be considered a systemic medication and the development of a new oral formulation was necessary beforehand [20].

SA001 is the prodrug of rebamipide with an ester pro-moiety developed by Samjin Pharmaceuticals Co., Ltd. (Seoul, Korea). In preclinical studies, SA001 was more effective than the rebamipide ophthalmic solution in stimulating the secretion of mucins from goblet cells (data on file). Moreover, it was shown to suppress pro-inflammatory cytokines such as IL-1, IL-6 and INF-γ and T cell activation in corneal epithelial cells and lachrymal glands (data on file). Therefore, with sufficiently high systemic exposure, SA001 was expected to reach the therapeutic concentrations of rebamipide at the target site and be a potential therapeutic for DED patients, especially with an autoimmune component. This study evaluated the safety, tolerability and pharmacokinetic characteristics of SA001 and rebamipide to investigate the suitability of this new oral formulation for future clinical trials.

## 2. Results

### 2.1. Study Population

A total of 96 healthy subjects were enrolled and completed the study (40 subjects in Trial A, 32 subjects in the SAD cohort of Trial B and 24 subjects in the MAD cohort of Trial B). All subjects in Trial A completed the study, and two subjects in Trial B (one subject from the 720 mg dose group in the SAD and MAD cohort each, respectively) withdrew their consents after the administration of SA001. The safety analysis was conducted in 96 subjects (40 subjects in Trial A, 32 subjects in the SAD study of Trial B and 24 subjects in the MAD study of Trial B) who received treatment at least once. The PK analysis was conducted in 94 subjects (40 subjects in Trial A, 31 subjects in the SAD cohort and 23 subjects in the MAD cohort of Trial B). Demographic characteristics between the dose groups were not statistically different in both Trial A and Trial B.

### 2.2. Pharmacokinetics

Both SA001 and its active metabolite, rebamipide, showed a mono-exponential decrease pattern until 12 h post dose while there was a sign of a decrease in the terminal slope of rebamipide after 12 h post-dose (Figure 1, Appendix A). SA001 was rapidly absorbed with a median T_max_ ranging from 0.4 to 1 h. SA001 was then rapidly converted to rebamipide which showed a T_max_ ranging from 0.5 to 1 h. The t_1/2_ of rebamipide showed large inter-individual variability and ranged from 1.85 to 21.72 h. However, the mean MRT of rebamipide was similar among the different dose groups ranging from 2.35 to 3.77 h (Table 1). The CL_R_ of SA001 was negligible compared to the overall clearance with the fraction of urine excreted ranging from 0 to 0.0023. The overall concentration-time profiles after the multiple doses in SA001 and rebamipide were similar to those after the single dose with a mono-exponential decrease pattern until 12 h post-dose (Figure 1, Appendix A). The accumulation ratio of rebamipide after multiple doses was similar across the various dose groups and dose intervals (b.i.d. or t.i.d.) ranging from 1.14 to 1.26 (Table 2). The absorption of rebamipide was delayed in the fed state, and the median T_max_ of rebamipide increased from 0.5 to 2.5 h in the 120 mg dose group and 1 to 2 h in the 300 mg dose group (Appendix A). There was little effect of food on the AUC_last_ of rebamipide, and the C_max_ after the fasted state was approximately 50% higher compared to the fed state (Appendix A). The overall concentration of rebamipide was higher after the administration of SA001 compared to the conventional rebamipide of a similar dose (Figure 2). The dose-adjusted AUC_last_ and C_max_ of rebamipide after the SA001 administration was 2.20 and 5.45 times higher in the 240 mg dose group and 4.73 and 11.94 times higher in the 600 mg dose group compared to the conventional rebamipide, respectively (Table 3). The 95% confidence interval (CI) of the slope in the power model was 1.0 (0.91–1.18 for AUC_last_, 0.94–1.20 for C_max_, 0.80–1.35 for AUC within a dosing interval at steady state (AUC_τ,ss_) and 0.85–1.31 for C_max_ at steady state (C_max,ss_)) for rebamipide and showed dose-proportional PK characteristics.

### 2.3. Safety and Tolerability

SA001 was well tolerated after the single and multiple administrations. No serious AEs were reported. All the AEs were mild, and all subjects recovered from them without any complications. Ten AEs from five subjects in the SAD study and 14 AEs from eight subjects in the MAD study were reported after the administration of SA001 (Table 4, Appendix A). Among them, seven AEs from five subjects in the SAD study and 11 AEs from seven subjects in the MAD study were considered to be related to the treatments. The linear model investigating the relationship between the plasma concentration of SA001 and rebamipide with the ΔΔ QTcF showed that neither of them provoked QT prolongation (Appendix A). No clinically significant changes were found in the vital signs and clinical laboratory tests.

## 3. Discussion

Through these two studies, the PK characteristics of SA001 were thoroughly investigated. SA001 was rapidly absorbed and converted to rebamipide. The apparent clearance of SA001 decreased as the dose increased from 60 to 1080 mg in the SAD study. Considering that the apparent volume of the distribution also decreased in the higher doses, this tendency could be explained by the increase in bioavailability due to saturation of the first pass effect. The AUC_last_ of SA001 increased more dose-proportionally and could have contributed to the decreasing tendency in the metabolic ratio of rebamipide as the dose was increased. The t_1/2_ of SA001 was less than 1 h and was undetectable in most subjects 8 h post-dose. On the other hand, the t_1/2_ of rebamipide was longer than SA001 and was elongated in the terminal phase, implying multi-compartmental PK characteristics.

When compared to the conventional rebamipide oral formulation, the systemic exposure of rebamipide was higher in SA001. Rebamipide is classified as BCS class IV [21], and numerous attempts were made to improve its absorption [22,23,24]. Developing prodrugs to enhance bioavailability and efficacy has been a strategy in various medical fields [25,26,27], and SA001 successfully enhanced the systemic exposure of rebamipide. Rebamipide is known for not only cytoprotection but also wound healing and anti-inflammatory effects, and its effects are expected to be universal in various tissues [7]. The poor absorption and low systemic exposure of the conventional oral formulation of rebamipide was the major hurdle for its application to other diseases. Evaluation of the PK characteristics with safety and tolerability profiles in a wide dose range and better exposure of rebamipide observed in SA001 would be useful for exploring rebamipide for other indications of interest. In fact, ongoing phase 2 clinical trials with SA001 are not only limited to DED patients (NCT03723798) but also to patients with primary Sjogren’s syndrome (NCT05269810). Considering the PK characteristics and the preclinical efficacy model, SA001 was planned to be administered only twice daily in the described trials and it is expected that patient compliance could be improved compared to the previous eye drop solutions. Moreover, the absorption of the conventional rebamipide was known to be restricted at higher doses, and the AUC failed to increase dose-proportionally in a previous study [28]. Likewise, in our study, the AUC_last_ in the 600 mg dose group for the conventional rebamipide was only 1.7 times higher than that of the 200 mg dose group. In contrast, the PK characteristics of rebamipide were dose proportional after the administration of SA001. This could be a great advantage when designing further clinical trials.

SA001 was tolerable in all the dose groups. Subjects were administered up to 1080 mg of SA001 in the SAD study and 360 mg b.i.d. or 240 mg t.i.d. in the MAD study and had no complications. No particular AEs occurred repeatedly among the subjects, and there was no significant finding in the laboratory tests and vital signs. There was no subject who could not complete the study schedule due to a safety issue, which was comparable with the safety and tolerability profiles of the conventional rebamipide reported in previous studies [29,30]. Additionally, the higher plasma concentration of SA001 and rebamipide was not associated with QT prolongation. These safety and tolerability profiles combined with its favorable PK characteristics support further clinical development of SA001 as a candidate for anti-inflammatory agents and for other indications when combined with other preclinical efficacy studies.

## 4. Materials and Methods

This study was approved by the Korean Ministry of Food and Drug Safety (MFDS). Study documents were reviewed by the Institutional Review Board (IRB) of Seoul National University Hospital. The clinical trials were conducted in accordance with the Declaration of Helsinki and Korean Good Clinical Practice (KGCP) (ClinicalTrials.gov identifier: NCT02470286, NCT05303961).

### 4.1. Study Population

Healthy Korean male subjects aged between 19 and 45 years with a body mass index (BMI) of 18.0 to 27.0 were enrolled. Previous medical and surgical history, physical examination, vital signs, 12-lead electrocardiography (ECG) and clinical laboratory tests were evaluated. Subjects with a corrected QT interval by Fredericia (QTcF) exceeding 430 ms, a PR interval more than 200 ms or less than 110 ms and a QRS higher than 120 ms were excluded. Subjects with laboratory tests of AST/ALT exceeding 1.5 times the upper limits of normal or a glomerular filtration rate lower than 90 mL/min/1.73 m^2^ estimated by the Modification of Diet in Renal Disease (MDRD) equation were also excluded.

### 4.2. Study Design

This study was a dose-block, randomized, double-blind, placebo-controlled, single and multiple ascending dose study consisting of two trials (Appendix A). As this first in human study was an exploratory part of clinical development, the number of subjects was not calculated based on statistical power. Empirically, 8–10 subjects were enrolled per dose group in previous studies. In this study, eight subjects were randomized to either SA001 or placebo at a 3:1 ratio in each dose group. The initial dose was determined by calculating the maximum recommended starting dose (MRSD) from the US FDA guideline [31]. The no observed adverse event level (NOAEL) observed from repeated dose toxicity tests over 4 weeks in mice and beagles was 167 and 150 mg/kg, respectively. The MRSD was calculated by converting the NOAELs to the human equivalent doses (HEDs) and applying the average adult weight (60 kg) and the safety factor (=10). The MRSD in mice and beagles were 78 mg and 486 mg, respectively [32]. Based on these results, 60 mg was selected as the initial dose.

Trial A (SJSA001, NCT02470286) was a single ascending dose (SAD) study ranging from 60 mg to five-fold of the initial dose using the modified Fibonacci method. All subjects received an assigned single dose of SA001 or placebo in the fasted state with 150 mL of water, and food intake was prohibited until 4 h after the administration. In the 120 and 300 mg dose groups, the PK characteristics in the fed state were additionally evaluated. Blood samples for the PK analysis were collected at 0 (before dose), 0.25, 0.5, 1, 1.5, 2, 2.5, 3, 4, 5, 6, 8, 12, 24, 32 and 48 h and urine samples at 0 (before dose), 0–4, 4–8, 8–12, 12–24, 24–32 and 32–48 h post-dose.

The PK profiles of SA001 from Trial A and Trial B (SJSA001_02, NCT05303961) were additionally arranged. Trial B consisted of a SAD cohort and a multiple ascending dose (MAD) cohort (Appendix A). To select the appropriate doses, articles on rebamipide were explored [29,33,34,35]. The dose of rebamipide in SA001 was calculated based on the assumption that all the SA001 malonic acid salt (molecular weight (MW) of 587.17 g/mol) would be metabolized to rebamipide (MW of 370.70 g/mol). In one study, 900 mg/day of rebamipide was administered to investigate the gastric mucoprotective effect [36]. Therefore, 1080 mg of SA001, considered equivalent to 829 mg of rebamipide based on the ratio of the MW described above, was selected as the maximum dose. Consequently, 240, 480, 720 and 1080 mg of SA001 were administered in the SAD cohort. A common therapeutic dosage of rebamipide for peptic ulcer was 300 mg/day, and the PK characteristics of 240 and 720 mg of SA001 were compared with 200 and 600 mg of the conventional rebamipide oral formulation (Bamedin^®^, Samjin Pharmaceuticals Co., Ltd., Korea). In the MAD cohort, 360, 720 and 1080 mg of SA001 were chosen as the doses after reviewing the PK profiles of rebamipide and considering the possible dose regimen for future clinical trials. During the total 16 days of administration during the MAD cohort, subjects received half the assigned dose twice a day (b.i.d.) in the first 8 days (Period 1) and one-third of the assigned dose three times a day (t.i.d.) in the latter 8 days (Period 2) in the fasted state with 250 mL of water. For example, subjects received 180 mg of SA001 b.i.d. during period 1 and 120 mg of SA001 t.i.d. during period 2 for the 360 mg dose group. Blood samples were collected at 0 (before dose), 0.25, 0.5, 1, 1.5, 2, 2.5, 3, 4, 5, 6, 8, 12, 24, 32 and 48 h and urine samples at 0 (before dose), 0–4, 4–8, 8–12, 12–24, 24–32 and 32–48 h post dose in the SAD cohort. In the MAD cohort, blood samples were collected at 0 (before dose), 0.25, 0.5, 1, 1.5, 2, 2.5, 3, 4, 5, 6, 8 and 12 h and urine samples at 0 (before dose), 0–4, 4–8 and 8–12 h after the first dose (Day 1), at steady state after the b.i.d. administration (Day 8) and at steady state after the t.i.d. administration (Day 16).

For each blood sample, 2 mL of blood collected in a sodium heparin vacutainer were mixed with 4 mL of DMF (N,N’-dimethylformamide) and centrifuged at 6000× *g* for 5 min. Three aliquots of supernatant were stored at −70 ℃ until analysis. The urine samples of each subject were collected in a polyethylene bag. At the end of each period, the bag was gently mixed, and 30 mL of urine was equally distributed in three conical tubes (10 mL each). Plasma and urine concentrations of SA001 and rebamipide were analyzed separately using a validated liquid chromatography (LC)–tandem mass spectrometry (MS/MS) system in the positive ionization mode. Data were obtained with a mass-to-charge ratio (*m*/*z*) of 484.1 → 114.2 for SA001, 488.2 → 114.2 for its internal standard (SA001-d4), 371.1 → 216.1 for rebamipide and 375.0 → 216.1 for its internal standard (rebamipide-d4). The accuracy and precision of the QC samples were 85–115% (80–120% for LLOQ).

### 4.3. Pharmacokinetic Evaluation

The PK parameters were estimated through non-compartmental methods using the Phoenix WinNonlin^®^ software version 6.4 (Pharsight Co., Mountain View, CA, USA) and included the maximum plasma concentration (C_max_), time to reach C_max_ (T_max_), area under the concentration-time curve (AUC) calculated by the linear-up/log-down trapezoidal method, half-life (t_1/2_), apparent total clearance (CL/F), renal clearance (CL_R_), apparent volume of distribution (Vz/F), mean residence time (MRT), accumulation ratio (R_AUC24h_) and metabolic ratio. R_AUC24h_ was calculated as AUC in 24 h (AUC24h) at a steady state divided by AUC24h after the first dose. AUC24h at steady state was calculated as 2 × AUC12h in period 1 (b.i.d.) and 3 × AUC8h in period 2 (t.i.d.). AUC24h after the first dose was assumed as 2 × AUC12h. The metabolic ratio of rebamipide was calculated by the ratio of AUC from 0 to the time of last measured concentration (AUC_last_) of rebamipide to AUC_last_ of SA001.

### 4.4. Safety and Tolerability Evaluation

Safety and tolerability were evaluated throughout the study based on adverse events (AEs), physical examinations, vital signs, 12-lead ECG and clinical laboratory tests. AEs were observed throughout the study, and the investigators assessed their relationship with the treatments. For each time point, triplicate ECGs were implemented, and the median value was used for the analysis. The time-matched difference of the QTcF compared to the baseline (ΔΔ QTcF) was evaluated to determine whether SA001 could induce QT prolongation.

### 4.5. Statistical Analysis

All the data were summarized using descriptive statistics. SAS software version 9.4 (SAS Institute Inc., Cary, NC, USA) was used for the statistical analysis. Dose proportionality or PK linearity was analyzed for the AUC and C_max_ of SA001 and rebamipide using a power model, in which natural log-transformed PK parameters were regressed on the natural log-transformed dose. The geometric mean ratio (GMR) of the AUC and C_max_ of SA001 and rebamipide in the fasted to the fed state were analyzed in Trial A. The GMR of the dose-adjusted AUC and C_max_ between SA001 and conventional rebamipide were evaluated in Trial B. To explore the possible impacts of SA001 on the cardiovascular system, the relationship between drug concentration and ΔΔ QTcF was evaluated by a linear model.

## 5. Conclusions

The AUC and C_max_ of rebamipide were dose-proportional after single and multiple oral doses of SA001. The systemic exposure of rebamipide after the administration of SA001 was higher than that of the conventional rebamipide oral formulation. SA001 showed excellent safety and tolerability profiles without any clinically significant AEs.

## Figures and Tables

**Figure 1 pharmaceuticals-16-00132-f001:**
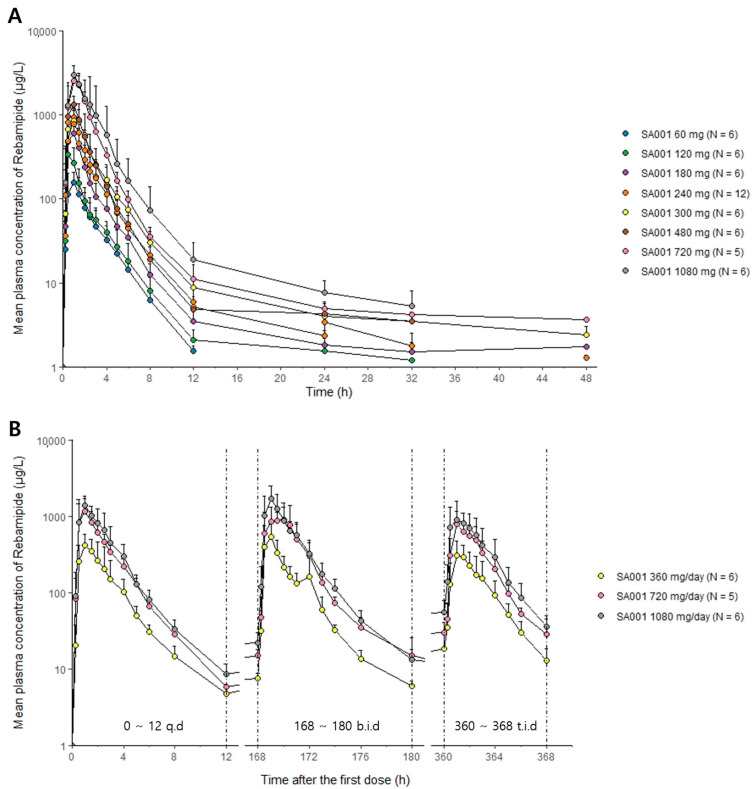
Mean plasma concentration-time profiles of rebamipide after (**A**) single and (**B**) multiple oral administration of SA001 in log-linear scale. SA001 was administered twice a day (b.i.d.) with half the assigned dose until 180 h after the first dose and three times a day (t.i.d.) with one-third of the assigned dose from 180 to 360 h after the first dose in the multiple ascending dose cohort. Bars represent standard deviations. Abbreviations: q.d.; quaque die (once a day).

**Figure 2 pharmaceuticals-16-00132-f002:**
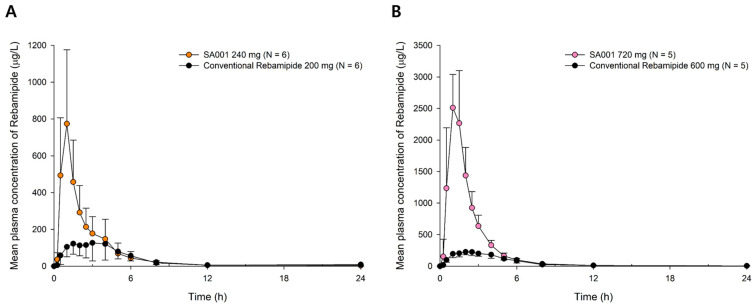
Mean plasma concentration-time profiles of rebamipide after single oral administration of SA001 and conventional rebamipide (Bamedin^®^) (**A**) SA001 240 mg and conventional rebamipide 200 mg (**B**) SA001 720 mg and conventional rebamipide 600 mg. Bars represent standard deviations.

**Table 1 pharmaceuticals-16-00132-t001:** Pharmacokinetic parameters of SA001 and rebamipide after single oral administration of SA001.

Parameter		SA001 60 mg (*N* = 6)	SA001 120 mg (*N* = 6)	SA001 180 mg (*N* = 6)	SA001 240 mg (*N* = 12)	SA001 300 mg (*N* = 6)	SA001 480 mg (*N* = 6)	SA001 720 mg (*N* = 5)	SA001 1080 mg (*N* = 6)
SA001	AUC_last_ (μg ∗ h/L)	5.76 ± 3.2	22.66 ± 11.69	59.12 ± 27.39	75.62 ± 39.13	98.81 ± 19.57	148.31 ± 42.56	348.64 ± 72.68	494.45 ± 128.20
	AUC_inf_ (μg ∗ h/L)	5.88 ± 3.21	22.91 ± 11.74	59.37 ± 27.35	75.97 ± 39.02	99.03 ± 19.71	148.80 ± 42.56	349.76 ± 72.80	495.44 ± 128.21
	C_max_ (μg/L)	10.17 ± 6.7	47.56 ± 28.89	90.97 ± 61.92	100.71 ± 55.14	127.28 ± 21.03	192.52 ± 114.76	357.11 ± 149.04	495.27 ± 127.46
	T_max_ (h)	0.4 (0.25, 1)	0.5 (0.25, 1.5)	0.5 (0.25, 1)	0.5 (0.25, 1)	0.5 (0.25, 1)	0.5 (0.5, 1)	1 (0.25, 1)	0.75 (0.5, 2)
	Vz/F (L)	11,222.46 ± 5808.18	8241 ± 7939.24	3826.44 ± 1925.96	4411.57 ± 2159.06	3681.79 ± 927.36	3931.61 ± 1094.99	2437.95 ± 628.76	2900.52 ± 572.15
	CL/F (L/h)	13,513.85 ± 7527.99	7430.12 ± 5249.91	3644.35 ± 1673.25	4032.10 ± 1969.11	3135.34 ± 645.44	3434.60 ± 901.37	2147.28 ± 542.00	2304.11 ± 584.09
	t_1/2_ (h)	0.59 ± 0.07	0.76 ± 0.25	0.72 ± 0.04	0.76 ± 0.89	0.84 ± 0.29	0.79 ± 0.08	0.80 ± 0.16	0.90 ± 0.22
	MRT_last_ (h) **	0.84 ± 0.31	0.83 ± 0.28	0.90 ± 0.21	0.97 ± 0.20	1.03 ± 0.38	1.00 ± 0.23	1.23 ± 0.27	1.31 ± 0.40
Rebamipide	AUC_last_ (μg ∗ h/L)	378.39 ± 142.89	591.41 ± 265.58	1203.37 ± 404.34	1725.48 ± 745.67	2352.47 ± 700.48	2485.45 ± 358.24	5331.31 ± 1022.09	6705.68 ± 3193.37
	AUC_inf_ (μg ∗ h/L)	382.6 ± 142.39	599.48 ± 264.13	1227.52 ± 410.15	1743.88 ± 746.05	2436.55 ± 731.33	2498.92 ± 357.54	5399.58 ± 1078.78	6768.84 ± 3215.15
	C_max_ (μg/L)	175.95 ± 50.12	344.94 ± 209.99	701.67 ± 184.94	958.11 ± 449.01	1148.74 ± 286.65	1415.42 ± 290.40	2772.00 ± 452.31	3458.74 ± 715.20
	T_max_ (h)	0.75 (0.5, 1)	0.5 (0.5, 1)	1 (0.5, 1)	1 (0.5, 1)	1 (0.5, 1.5)	0.5 (0.5, 1)	1 (0.5, 1.5)	1 (1, 2.5)
	t_1/2_ (h)	1.85 ± 0.25	3.68 ± 2.13	10.56 ± 11.63	6.79 ± 5.96 *	21.72 ± 13.03	2.27 ± 0.60	11.94 ± 13.56	7.72 ± 5.30
	MRT_last_ (h)	2.68 ± 0.40	2.92 ± 0.95	2.74 ± 0.56	2.80 ± 0.68	3.77 ± 0.58	2.35 ± 0.33	2.49 ± 0.45	2.62 ± 0.51
	Metabolic ratio ***	80.15 ± 41.66	28.44 ± 6.73	21.51 ± 3.64	25.35 ± 11.28	23.72 ± 4.76	17.33 ± 2.63	15.63 ± 3.20	13.47 ± 3.79

Notes: All values but T_max_ are presented as the arithmetic mean ± standard deviation. T_max_ values are represented as the median (minimum, maximum). * The value of one subject from Trial A was excluded because it was not estimable. ** MRT was calculated as follows: AUMC_last_/AUC_last_. *** Metabolic ratio was calculated as follows: AUC_last_ of rebamipide/AUC_last_ of SA001. Abbreviations: AUC_last_; AUC from 0 to the time of the last measured concentration, AUC_inf_; AUC from time of dosing extrapolated to infinity, C_max_; maximum plasma concentration, T_max_; time to reach maximum drug concentration, Vz/F; apparent volume of distribution, CL/F; apparent total clearance, t_1/2_; elimination half-life, MRT_last_; Mean residence time, CL_R_: renal clearance.

**Table 2 pharmaceuticals-16-00132-t002:** Pharmacokinetic parameters of SA001 and rebamipide at steady state after multiple oral administration of SA001.

	Parameter	SA001 360 mg (*N* = 6)		SA001 720 mg (*N* = 5)		SA001 1080 mg (*N* = 6)	
		180 mg b.i.d.	120 mg t.i.d.	360 mg b.i.d.	240 mg t.i.d.	540 mg b.i.d.	360 mg t.i.d.
SA001	AUC24h after the first dose (μg ∗ h/L) *	70.18 ± 43.12		300.06 ± 137.61		352.14 ± 177.65	
	AUC24h at steady state (μg ∗ h/L) *	83.92 ± 26.95	76.23 ± 27.91	294.14 ± 134.65	285.86 ± 75.46	393.09 ± 125.71	456.06 ± 155.06
	C_max,ss_ (μg/L)	53.71 ± 33.34	25.63 ± 13.65	160.82 ± 70.45	106.61 ± 27.65	226.03 ± 133.92	155.69 ± 50.43
	T_max,ss_ (h)	0.5 (0.5, 4)	0.8 (0.3, 3.0)	0.5 (0.5, 2.03)	0.5 (0.5, 1.77)	0.75 (0.5, 3.0)	0.5 (0.5, 1.5)
	R_AUC24h_ *	1.40 ± 0.54	1.23 ± 0.44	1.03 ± 0.28	1.09 ± 0.51	1.21 ± 0.30	1.49 ± 0.73
Rebamipide	AUC24h after the first dose (μg ∗ h/L) *	2136.39 ± 914.52		5235.66 ± 1682.13		6415.98 ± 1866.11	
	AUC24h at steady state (μg ∗ h/L) *	2344.83 ± 629.89	2590.34 ± 877.18	5905.92 ± 2007.78	5961.95 ± 1456.02	7616.60 ± 2222.60	7978.21 ± 2332.03
	C_max,ss_ (μg/L)	626.63 ± 222.76	380.12 ± 117.92	1266.13 ± 454.17	891.05 ± 255.37	2001.88 ± 616.93	1375.22 ± 309.73
	T_max,ss_ (h)	1.0 (0.5, 4.0)	1.25 (1.0, 3.02)	1.5 (1.0, 2.03)	1.0 (1.0, 1.77)	1.0 (0.5, 3.0)	1.0 (0.5, 2.03)
	R_AUC24h_ *	1.18 ± 0.28	1.26 ± 0.19	1.14 ± 0.19	1.23 ± 0.52	1.21 ± 0.28	1.25 ± 0.18

Notes: All values but T_max_ are presented as the arithmetic mean ± standard deviation. T_max_ values are represented as the median (minimum, maximum). * Accumulation ratio of AUC24h was calculated by the following equations: (2 × AUC12h at steady state)/(2 × AUC12h after the first dose) at period 1 (b.i.d. administration) and (3 × AUC8h at steady state)/(2 × AUC12h after the first dose) at period 2 (t.i.d. administration). Assumed that AUC_0–12h_ ≒ AUC_12–24h_ in period 1 and AUC_8h_ ≒ AUC_8–16h_ ≒ AUC_16–24h_ in period 2. Abbreviations: AUC24h; AUC in 24 h, C_max,ss_; maximum plasma concentration at steady state, T_max,ss_; time to reach maximum drug concentration at steady state, R_AUC24h_; Accumulation ratio based on AUC 24 h.

**Table 3 pharmaceuticals-16-00132-t003:** Pharmacokinetic parameters of rebamipide after single oral administration of SA001 and conventional rebamipide (Bamedin^®^).

Parameter	SA001 240 mg (*N* = 6)	Conventional Rebamipide 200 mg (*N* = 6)	GMR * (90% CI)	SA001 720 mg (*N* = 5)	Conventional Rebamipide 600 mg (*N* = 5)	GMR * (90% CI)
AUC_last_ (μg ∗ h/L)	1516.87 ± 697.51	733.85 ± 344.36		5331.31 ± 1022.09	1220.53 ± 212.86	
AUC_last_/D (μg ∗ h/L/mg) **	8.23 ± 3.79	3.67 ± 1.72	2.20 (1.73–2.80)	9.64 ± 1.85	2.03 ± 0.35	4.73 (3.46–6.46)
C_max_ (μg/L)	785.68 ± 393.39	158.73 ± 87.37		2772.00 ± 452.31	252.82 ± 46.06	
C_max_/D (μg/L/mg) **	4.26 ± 2.13	0.79 ± 0.44	5.45 (4.22–7.03)	5.01 ± 0.82	0.42 ± 0.08	11.94 (8.73–16.32)

Notes: All values are presented as the arithmetic mean ± SD. * Geometric mean ratio of SA001 to conventional rebamipide. ** Calculated assuming that SA001 240 mg was equivalent to 184.27 mg of rebamipide, and SA001 720 mg was equivalent to 552.81 mg of rebamipide. Abbreviations: AUC_last_; AUC from 0 to the time of the last measured concentration, C_max_; maximum plasma concentration, D: Dose.

**Table 4 pharmaceuticals-16-00132-t004:** Summary of treatment-emergent adverse events (TEAEs) by systemic organ class and preferred term after multiple oral administrations of SA001.

System Organ Class Preferred Term	Placebo (*N* = 6)	SA001 360 mg (*N* = 6)	SA001 720 mg (*N* = 6)	SA001 1080 mg (*N* = 6)	Total (*N* = 24)
Subjects with TEAEs	0	1 (16.7) (2)	6 (100.0) (9)	1 (16.7) (3)	8 (33.3) (14)
Gastrointestinal disorders	0	1 (16.7) (2)	0	0	1 (4.2) (2)
Nausea	0	1 (16.7) (1)	0	0	1 (4.2) (1)
Vomiting	0	1 (16.7) (1)	0	0	1 (4.2) (1)
General disorders and administration site conditions	0	0	1 (16.7) (1)	0	1 (4.2) (1)
Catheter site erythema	0	0	1 (16.7) (1)	0	1 (4.2) (1)
Infections and infestations	0	0	1 (16.7) (1)	0	1 (4.2) (1)
Oral pustule	0	0	1 (16.7) (1)	0	1 (4.2) (1)
Musculoskeletal and connective tissue disorders	0	0	1 (16.7) (1)	0	1 (4.2) (1)
Myalgia	0	0	1 (16.7) (1]	0	1 (4.2) (1)
Respiratory, thoracic and mediastinal disorders	0	0	4 (66.7) (5)	1 (16.7) (2)	5 (20.8) (7)
Dyspnea	0	0	1 (16.7) (1)	0	1 (4.2) (1)
Epistaxis	0	0	1 (16.7) (1)	0	1 (4.2) (1)
Nasal congestion	0	0	1 (16.7) (1)	1 (16.7) (1)	2 (8.3) (2)
Rhinorrhea	0	0	2 (33.3) (2)	1 (16.7) (1)	3 (12.5) (3)
Skin and subcutaneous tissue disorders	0	0	1 (16.7) (1)	1 (16.7) (1)	2 (8.3) (2)
Papule	0	0	1 (16.7) (1)	0	1 (4.2) (1)
Erythema	0	0	0	1 (16.7) (1)	1 (4.2) (1)

Notes: TEAEs are displayed as the number of subjects (percentage of subjects) <number of events>.

## Data Availability

Data is contained within the article and Supplementary Material.

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
