# Peer review of "Evaluation of Safety, Tolerability and Pharmacokinetic Characteristics of SA001 and Its Active Metabolite Rebamipide after Single and Multiple Oral Administration"

_pharmaceuticals, 2023, doi:10.3390/ph16010132_

Round 1

Reviewer 1 Report

A most impressive study, comprehensive in it's design, with the important benchmarks for such work all highlighted. It was a delight to read, and will provide enhanced knowledge to our field. 

Author Response

We would like to thank the Reviewers for their comments.

Reviewer 2 Report

The manuscript is with merit and the result are worth reporting but 

the authors should provide:

- statistical power statement or at least some justification of the study sample n

- the methods if possible per the journal guidelines should be moved after the introduction

- in the discussion the authors should provide an insight of future research in this field

Reviewer 3 Report

The paper entitled “First in human evaluation of safety, tolerability and pharmaco-kinetic characteristics of SA001 and its active metabolite rebamipide after single and multiple oral administration in healthy male subjects” is a study based on a new oral formulation of rebamipide for dry eye disease (DED).

The aim of this study was to evaluate the tolerability, safety, and pharmacokinetic characteristics of SA001 and rebamipide and to assess the suitability of this new oral formulation for future clinical trials.

The title does not need to be a mini-abstract and could be shortened and improved to read “i.e., Evaluation of…administration”.

The authors should include a brief definition of “rebamipide” in the abstract considering that most clinicians that read the abstract do not know these terms, and probably will not be enticed to read the paper unless the definition is provided.  

Editing can improve the English and flow of the text. The authors should include a brief description and comparison regarding protective agents currently used today for patients with DED, and explain if and how this substance and the new formulation can be an improvement in therapy for DED in the Discussion section.

Clinical perspectives and implications are lacking in the Discussion and should be included.  The authors should also comment on the duration of the effects of rebamipide on clinical signs and symptoms.

The study adds to the literature by validating the potential benefits of new formulations of this substance and can help in planning future models regarding potential protective agents for DED. The main question of the manuscript regarding safety and tolerability issues has been properly addressed. The results show that SAA001 showed good safety and tolerability profiles without any clinically significant adverse effects.

The study has been correctly planned and represents a valid model for future studies in this field. It is nicely written and of potential interest. The study provides objective results and is relevant in this field. The conclusions are consistent with the continents presented throughout the text and the main questions have been addressed in an appropriate manner. References are appropriate. The figures and tables are pertinent, and descriptive and assist in describing the results.
